# Public Willingness and Determinants of COVID-19 Vaccination at the Initial Stage of Mass Vaccination in China

**DOI:** 10.3390/vaccines9101172

**Published:** 2021-10-13

**Authors:** Yi-Miao Zhao, Lin Liu, Jie Sun, Wei Yan, Kai Yuan, Yong-Bo Zheng, Zheng-An Lu, Lin Liu, Shu-Yu Ni, Si-Zhen Su, Xi-Mei Zhu, Na Zeng, Yi-Miao Gong, Ping Wu, Mao-Sheng Ran, Yue Leng, Jie Shi, Le Shi, Lin Lu, Yan-Ping Bao

**Affiliations:** 1National Institute on Drug Dependence and Beijing Key Laboratory of Drug Dependence, Peking University, Beijing 100191, China; 2011210099@stu.pku.edu.cn (Y.-M.Z.); 1510306117@pku.edu.cn (L.L.); nishuyu@pku.edu.cn (S.-Y.N.); zengna2017@163.com (N.Z.); wuping@bjmu.edu.cn (P.W.); shijie@bjmu.edu.cn (J.S.); 2School of Public Health, Peking University, Beijing 100191, China; 3Peking University Sixth Hospital, Peking University Institute of Mental Health, NHC Key Laboratory of Mental Health (Peking University), National Clinical Research Center for Mental Disorders (Peking University Sixth Hospital), Peking University, Beijing 100191, China; linliu@bjmu.edu.cn (L.L.); weiyan@bjmu.edu.cn (W.Y.); yuankai@pku.edu.cn (K.Y.); yongbozheng@bjmu.edu.cn (Y.-B.Z.); 1911210616@bjmu.edu.cn (Z.-A.L.); sizhensu@bjmu.edu.cn (S.-Z.S.); ximeizhu@bjmu.edu.cn (X.-M.Z.); 1801111460@pku.edu.cn (Y.-M.G.); 4Pain Medicine Center, Peking University Third Hospital, Beijing 100191, China; sunjie0615@pku.edu.cn; 5Peking-Tsinghua Center for Life Sciences and PKU-IDG/McGovern Institute for Brain Research, Peking University, Beijing 100871, China; 6Beijing Friendship Hospital, Capital Medical University, Beijing 100050, China; 7Department of Social Work and Social Administration, University of Hong Kong, Hong Kong, China; msran@hku.hk; 8Department of Psychiatry and Behavioral Sciences, University of California, San Francisco, CA 94121, USA; Yue.Leng@ucsf.edu

**Keywords:** COVID-19 vaccine, vaccine hesitancy, vaccine resistance, determinants

## Abstract

The present study assessed the willingness of the general population to receive COVID-19 vaccines and identified factors that influence vaccine hesitancy and resistance. A national online survey was conducted from 29 January 2021 to 26 April 2021 in China. Multinomial logistic regression analyses were conducted to identify factors that influence vaccine hesitancy and resistance. Of the 34,041 participants surveyed, 18,810 (55.3%) were willing to get vaccinated, 13,736 (40.3%) were hesitant, and 1495 (4.4%) were resistant. Rates of vaccine acceptance increased over time, with geographical discrepancies in vaccine hesitancy and resistance between provinces in China. Vaccine safety was the greatest concern expressed by most participants (24,461 [71.9%]), and the major reason for participants’ refusing vaccination (974 [65.2%]). Government agencies (23,131 [68.0%]) and social media (20,967 [61.6%]) were the main sources of COVID-19 vaccine information. Compared with vaccination acceptance, female, young and middle-aged, high income, and perceived low-risk of infection were associated with vaccine hesitancy. Histories of allergic reactions to other vaccines and depression symptoms were related to vaccine resistance. Common factors that influenced vaccine hesitancy and resistance were residing in cities and perceiving less protection with vaccines than with other protective measures. The results indicate that the rate of vaccine resistance is relatively low, but vaccine hesitancy is common. Individuals who are female, young and middle-aged, with a high income, and residing in cities are more likely to be hesitant for vaccination and should be the target populations for vaccination campaigns. Specific vaccine messaging from the government and social media could alleviate public concerns about vaccine safety and efficacy.

## 1. Introduction

Coronavirus disease 2019 (COVID-19) has resulted in more than 200 million confirmed cases with 4 million deaths in more than 200 countries and regions as of August 2021 [1]. In the absence of an effective therapy, vaccination against COVID-19 has been regarded as one of the most cost-effective ways to prevent and control the pandemic. Numerous governments around the world are accelerating COVID-19 vaccine research and development. According to the World Health Organization (WHO) COVID-19 vaccine tracker and landscape [2], as of 22 August 2021, there have been 296 vaccine candidates, 112 of which reached the clinical phase. However, unequal vaccine distribution has been found globally, with much of the current supply directed toward high-income countries. Although the WHO has listed 14 vaccines for emergency use [3], COVID-19 vaccine supplies remain inadequate for the vast majority of low- and middle-income countries [4]. Establishing successful COVID-19 vaccination programs in low- and middle-income countries is key for controlling the pandemic worldwide with extensive geographic and population coverage [5].

Having licensed vaccines is not sufficient to end the COVID-19 pandemic, but it is crucial to ensure enough people get vaccinated to achieve herd immunity [4]. The first Chinese inactivated vaccine was developed by state-owned Sinopharm. It was conditionally approved on 31 December 2020. Although the COVID-19 vaccine was available for free, the number of people who got vaccinated was limited in the early stage of the vaccination program in China with vaccine skepticism. Using a pooled estimate of the COVID-19 R_0_ of 2.2–2.7 for China [6] and assuming a best-case scenario in which a vaccine has perfect efficacy, this yields a projection that at least 55% of the population would need to be vaccinated to achieve herd immunity. Despite the fact that having been vaccinated with more than 1.9 billion doses as of 22 August 2021, the emergence of highly contagious Dela variants indicated that vaccine booster shots are still needed for the Chinese population, demonstrating that there is a long way to go to achieve herd immunity.

Vaccine hesitancy is a major cause of the low rate of vaccine uptake, and has been identified as one of the top 10 global health threats in 2019 [7,8]. The SAGE Working Group on Vaccine Hesitancy defined vaccine hesitancy as a delay in the acceptance or refusal of vaccination despite the availability of vaccination services [8]. Previous studies of COVID-19 vaccine willingness reported substantial variation both across and within countries [5]. A global survey of 19 countries indicated differences in vaccine acceptance rates, ranging from nearly 90% in China to less than 55% in Russia [9]. In China, the proportion of vaccine acceptance declined from 91.9% in March 2020 (severe phase) to 88.6% in November–December 2020 [10]. However, most existing studies were conducted immediately before the COVID-19 vaccine was conditionally approved. Most previous studies assessed vaccination willingness and related factors when COVID-19 vaccines were unavailable [5,11,12,13,14,15,16]. Understanding the factors that drive COVID-19 vaccine acceptance and exploring possible reasons for vaccine hesitancy in settings where the vaccine is available are important for effective control of the COVID-19 pandemic worldwide, especially in low- and middle-income countries. Public confidence in vaccination was shown to be impacted by negative vaccine information, including adverse effects after actual immunization. Additionally, with the coming of the so-called “regular epidemic prevention and control” phase and the possible emergence of new variants in many countries, public fear of the pandemic and the perceived risk of infection could be changed, which may lead to the changes in vaccination willingness [10].

Identifying target populations who opposed or were hesitant to receive vaccination and understanding why these people were less willing to be vaccinated may contribute to tailored immunization programs and vaccination campaign success. Thus, we conducted the present study from 29 January to 26 April 2021 when the Chinese government had just started to provide the general population with COVID-19 vaccines. We sought to assess vaccination willingness among the general population across time and regions in China. Furthermore, we explored the reasons for refusing vaccination and the factors influencing vaccine hesitancy and resistance.

## 2. Methods

### 2.1. Study Design and Participants

This study received approval from the ethics committee of Peking University Sixth Hospital (Institute of Mental Health). Informed consent was received online before the respondents began the questionnaire. This study followed American Association for Public Opinion Research (AAPOR) reporting guidelines and the Strengthening the Reporting of Observational Studies in Epidemiology (STROBE) statement.

This cross-sectional online study was conducted from 29 January 2021 to 26 April 2021. A self-report questionnaire was designed to investigate the willingness for COVID-19 vaccination among the general Chinese population. It was administered through the health page on the Chinese website Joybuy, a large e-commerce and information service platform that provides online health products and services in China [17].

### 2.2. Measurements and Covariates

#### 2.2.1. Willingness of COVID-19 Vaccination

Participants were asked, “Would you accept a new vaccine to prevent COVID-19?” They were classified as “vaccine acceptance” if they responded “have been vaccinated,” “hope to receive vaccination as soon as possible,” or “not recommended by the current guideline, but hope to receive vaccination in the future.” They were classified as “vaccine hesitant” if they responded “delay vaccination until confirming vaccine safety and efficacy.” These were classified as “vaccine resistant” if they responded “refuse.”

#### 2.2.2. Sociodemographic Characteristics, Epidemic-Related Factors, Vaccination-Specific Factors, and Health Status

Sociodemographic characteristics included sex, age, geographic region, living area, education, monthly family income, and marital status. Epidemic-related factors included COVID-19 infection status, frontline worker, local rebound of COVID-19, quarantine experience, and perceived effectiveness of protective measures, including vaccination, social distancing, wearing a mask, and maintaining personal hygiene. The perceived effectiveness of protective measures was measured using visual analog scales (VASs). Scores ranged from 0 (totally ineffective) to 10 (totally effective) and were classified as “vaccination can offer less protection than other protective measures” if VAS scores of perceived effectiveness of vaccination were lower than average scores of other protective measures. Vaccination-specific factors included access to COVID-19 vaccine information, contents and sources of vaccine information, reasons for refusing vaccination, attitudes about adverse effects after immunization, history of allergic reactions to other vaccines, and history of receiving influenza vaccines. Health status included a history of chronic diseases, a history of psychological disorders, a family history of psychological disorders, depression symptoms measured by the Patient Health Questionaire-9 (PHQ-9), and anxiety symptoms measured by the Generalized Anxiety Disorder-7 (GAD-7). Total scores of the two scales were interpreted as the following: normal (0–4), mild (5–9), and moderate to severe (10–27) depression symptoms for the PHQ-9 [18], and normal (0–4), mild (5–9), and moderate to severe (10–21) anxiety symptoms for the GAD-7 [19].

### 2.3. Statistical Analysis

Participants’ characteristics were summarized using frequencies and percentages. *χ*^2^ tests were conducted with associated *p* values for these three sets of comparisons: vaccine acceptance, vaccine hesitancy and vaccine resistance. One-way analysis of variance (ANOVA) and *χ*^2^ tests were performed to compare continuous and categorical variables, respectively, among the vaccine acceptance, vaccine hesitancy, and vaccine resistance groups. Multinomial logistic regression analyses were performed to calculate adjusted odds ratios (AORs) and 95% confidence intervals (CIs) of potential factors that influence vaccine hesitancy and resistance, including sociodemographic characteristics, epidemic-related factors, vaccination-specific factors, and health status, with the vaccine acceptance group set as the reference category. Spatial data analyses were conducted using ArcGIS 10.7 (ESRI Corp., Redlands, CA, USA). The other analyses were conducted using SPSS 23 software. Two-tailed values of *p* < 0.05 were considered statistically significant.

## 3. Results

### 3.1. Sociodemographic Characteristics

In summary, 74,588 people clicked on the survey page, and 34,291 submitted the questionnaire voluntarily, with a participation rate of 46.0%. A total of 250 respondents who were younger than 18 years old were excluded because obtaining online informed consent from their parents was not realistic under the present conditions. Finally, 34,041 participants from 34 provinces in China were included in this study, with an effective response rate of 99.3%. More than half of the participants (17,396 [51.1%]) were recruited from March 1 to 31, 2021. Of the total sample, most of the participants were female (18,309 [53.8%]), 18–39 years old (20,7272 [60.9%]), in eastern China (13,321 [39.2%]) and urban areas (26,942 [79.1%]), with a college degree or higher (26,957 [79.2%]), married (26,392 [77.5%]), and with a 5000 RMB to 11,999 RMB monthly family income (15,961 [46.9%]). This survey included data from 1074 (3.2%) COVID-19 frontline workers and 104 (0.3%) individuals with confirmed or suspected cases of COVID-19. Table 1 provides other details of this sample.

### 3.2. Prevalence of Vaccine Acceptance, Hesitancy, and Resistance

Overall, 18,810 (55.3%) of the respondents were accepting of a COVID-19 vaccine, including 5103 (15.0%) participants who had been vaccinated, 9940 (29.2%) who hoped to receive vaccination as soon as possible, and 3767 (11.1%) who were not recommended by the current guideline but hoped to receive vaccination in the future. A total of 13,736 (40.3%) were hesitant about such a vaccine, and 1495 (4.4%) were resistant. Figure 1 shows the rates of vaccine acceptance, hesitancy, and resistance. The prevalence of vaccine acceptance was significantly higher among men and those aged 60 years or older. Women and those 18–39 years old had a significantly higher rate of vaccine hesitancy. Table 2 presents the prevalence of vaccination willingness among different population subgroups. The proportion of people who were willing to accept vaccination increased over time (from 49.8% in February 2021 to 71.4% in April 2021), whereas the number of participants who were hesitant about or resistant to vaccination decreased over time (Figure 2). With regard to differences in vaccination willingness among different geographical regions, the rate of vaccine acceptance was the highest in northeast regions (1362 [56.8%]). The rate of vaccine hesitancy was the highest in eastern (5419 [40.7%]) and central (1206 [40.7%]) regions. The rate of vaccine resistance was the highest in eastern regions (627 [4.7%]). Figure 3 shows the details of vaccination willingness in the different provinces.

### 3.3. Contents and Sources of COVID-19 Vaccine Information and Reasons for Refusing Vaccination

The content of greatest concern for the public was vaccine safety (24,461 [71.9%]), followed by vaccine efficacy (22,630 [66.5%]) and vaccine research and development progress (19,022 [55.9%]). Participants were less concerned about the vaccine delivery schedule (15,942 [46.8%]) and price (10,450 [30.7%]). Most participants (23,131 [68.0%]) received vaccine information from government agencies, followed by social media (20,967 [61.6%]). With regard to reasons why 1495 participants refused vaccination, the most commonly reported reason was concern about vaccine safety (974 [65.2%]). Worrying about vaccine quality (633 [42.3%]) and efficacy (564 [37.7%]) or the perceived low risk of COVID-19 infection (522 [34.9%]) were other common reasons for vaccine resistance. Only 91 participants (6.1%) refused vaccination because of having vaccine contraindications. Figure 4 presents detailed results of contents and sources of vaccine information and reasons for refusing vaccination.

### 3.4. Factors Associated with Vaccine Hesitancy and Resistance

Multinomial logistic regression analyses showed that those who were hesitant about vaccination compared with those who accepted vaccination were more likely to be female (AOR = 1.21 [95% CI: 1.16–1.27]), to be younger (18–39 years old vs. >60 years old, AOR = 1.67 [95% CI: 1.38–2.01]; 40–59 years old vs. >60 years old, AOR = 1.53 [95% CI: 1.27–1.84]), to have a higher income (compared with <5000 RMB; AOR = 1.07 [95% CI: 1.01–1.13] for 5000–19,999 RMB; AOR = 1.10 [95% CI: 1.03–1.17] for >12,000 RMB), and to perceive a lower risk of infection (AOR = 1.77 [95% CI: 1.41–2.20]). Participants who had a history of chronic diseases (AOR = 0.88 [95% CI: 0.81–0.96]) were less likely to be hesitant about vaccination (AOR = 0.88 [95% CI: 0.81–0.96]). With regard to vaccine resistance, individuals who had a history of allergic reactions to other vaccines (AOR = 1.45 [95% CI: 1.25–1.69]) and who had depression symptoms (AOR = 1.20 [95% CI: 1.06–1.36]) were more likely to refuse vaccination. Respondents who had a medium family monthly income (5000– 19,999 RMB, AOR = 0.81 [95% CI: 0.71–0.92]) and who experienced a local rebound of the epidemic (AOR = 0.84 [95% CI: 0.73–0.98]) were less likely to be vaccine resistant.

For common factors that influenced vaccine hesitancy and resistance, individuals who lived in cities (vaccine hesitancy, AOR = 1.13 [95% CI: 1.07–1.20]; vaccine resistance, AOR = 1.26 [95% CI: 1.09–1.45]) and who thought that vaccines offer less protection than other protective measures (vaccine hesitancy, AOR = 1.52 [95% CI: 1.45–1.60]; vaccine resistance, AOR = 2.22 [95% CI: 1.99–2.48]) were more likely to be unsure about or oppose vaccination. In contrast, respondents who were frontline workers (vaccine hesitancy, AOR = 0.43 [95% CI: 0.37–0.50]; vaccine resistance, AOR = 0.50 [95% CI: 0.34–0.73]), who proactively access COVID-19 vaccine information (vaccine hesitancy, AOR = 0.74 [95% CI: 0.69–0.79]; vaccine resistance, AOR = 0.22 [95% CI: 0.20–0.25]), who had a history of receiving influenza vaccines (vaccine hesitancy, AOR = 0.62 [95% CI: 0.59–0.65]; vaccine resistance, AOR = 0.40 [95% CI: 0.35–0.46]), and who experienced quarantine were less likely to have a higher probability of being resistant about or refusing vaccination. Over time, respondents were increasingly less inclined to be hesitant about vaccination (March 2021 vs. February 2021, AOR = 0.77 [95% CI: 0.73–0.81]; April 2021 vs. February 2021, AOR = 0.41 [95% CI: 0.38–0.45]) or oppose vaccination (March 2021 vs. February 2021, AOR = 0.84 [95% CI: 0.75–0.94]; April 2021 vs. February 2021, AOR = 0.43 [95% CI: 0.34–0.55]). Table 3 presents detailed results of the logistic regression analyses.

## 4. Discussion

This was a national online survey with large geographic coverage that assessed the willingness to receive COVID-19 vaccination among the general population in China when the Chinese government had just started to provide vaccines for free. More than half (55.3%) of the participants were willing to get vaccinated, while a proportion of the population remained hesitant about (40.3%) or opposed (4.4%) vaccination, with geographical differences among the 34 Chinese provinces. As mass vaccination progressed, the proportion of individuals who were hesitant about or opposed vaccination decreased, and more individuals were willing to accept vaccination against COVID-19. This survey found that women, young and middle-aged people, urban dwellers, and those with a higher income were more likely to be hesitant about vaccination. Thus, these could be target populations for education about vaccine safety and efficacy. Overall, these findings may help to identify key populations who are hesitant about or oppose vaccination and provide guidance on the design of tailored vaccination campaigns to promote vaccination acceptance in the general population.

Although it is not possible to directly compare the present results with other existing surveys because of differences in questionnaires and methodologies, our findings suggested that the proportion of the population who was willing to get vaccinated was higher when there was not an approved COVID-19 vaccine relative to post-vaccine approval periods. A national survey in the United States indicated that 10.8% of adults did not intend to receive vaccination, and 31.6% were unsure about vaccination during the severe epidemic phase [15]. Under an assumed scenario where there would be a vaccine available against COVID-19, a survey of a representative sample found that 26% of the population in the United Kingdom and 25% of the population in Ireland were vaccine hesitant, and 9% of the United Kingdom population and 6% of the Irish population were vaccine resistant [13]. Similarly, in a nationally representative sample of Australian parents, participants who were unsure about getting vaccinated or unwilling to get vaccinated had increased by 10.0% from April to June 2020 [20]. The observed decrease in the proportion of people who would accept a COVID-19 vaccine could be associated with the perception of a lower risk of COVID-19 infection and the perception of lower disease severity. Compared with western countries, the rate of vaccination acceptance is relatively high in China, partly because of the Chinese traditional culture of emphasizing collective interests rather than individual interests. However, given the minimal vaccination rate (55%) with regard to reaching herd immunity, as calculated above, it is not optimistic that less than 40% of the total respondents, which is far below the herd immunity threshold of 55%, had been vaccinated or were willing to get vaccinated as soon as possible.

Four sociodemographic factors were associated with vaccine hesitancy and resistance: female, young and middle-aged, residing in cities, and higher income. Women were more likely to be hesitant about vaccination, which is consistent with previous studies that identified gender-related difference in COVID-19 vaccine acceptance [12,21]. This can be partly explained by the fact that women of childbearing age or who are currently pregnant or breastfeeding may be more concerned about vaccine side effects, resulting in high rates of vaccine hesitancy [22]. Younger age was also related to vaccine hesitancy. Younger individuals may perceive themselves to have a low risk of getting COVID-19 or becoming a severe case [23]. For these people, highlighting the threat of suffering from COVID-19 is critical to reduce rates of vaccine hesitancy. Urban dwelling was also associated with vaccine resistance and hesitancy, which is worrisome when considering the high potential for community transmission in more densely populated areas. Respondents with a higher income were also more likely to be unsure about vaccination, which aligns with previous studies of other vaccines [24,25]. Similarly, rates of vaccine hesitancy and resistance were relatively high in more economically advanced eastern regions in China. People with a high income can acquire more vaccine information through social media, including vaccine misinformation, and thus may be more concerned about vaccine safety and efficacy [26]. Moreover, these people can afford various types of COVID-19 vaccines, which may make them more hesitant to receive the vaccine. In terms of health status, participants with depression symptoms were more likely to be resistant to vaccination, which may be explained by a higher probability of fear of adverse effects [27]. Therefore, promoting people’s mental health could increase the proportion of vaccine acceptance [28]. Numerous studies found a higher risk of morbidity and severe cases among participants with a history of chronic diseases [29,30], which may make them more inclined to be vaccinated because of concerns about COVID-19 infection.

Vaccine safety was of the greatest concern by the public, and most people refused vaccination because of this, which is consistent with numerous previous studies [12,15,16]. Adequate and comprehensive communications can build public trust in vaccines. Participants who proactively sought vaccine information had low rates of vaccine hesitancy and resistance. Therefore, the post-marketing surveillance of adverse events after immunization should be made public regularly, which will be critical to alleviate public concerns about vaccine safety [31].

COVID-19 vaccine efficacy was another factor of great concern to the public. If individuals thought that vaccines offered less protection than other protective measures, then they were more likely to be unsure about or oppose vaccination. Previous vaccination experiences also impacted their willingness to receive COVID-19 vaccination. For example, numerous studies showed that individuals with a history of receiving influenza vaccines were less likely to be vaccine hesitant and resistant [15,16,32]. Additionally, the negative role of experiencing allergic reactions to other vaccines in accepting vaccination was confirmed in the present study. The transparency of COVID-19 vaccine trials, the regulatory approval of vaccines, and post-marketing surveillance may boost public confidence in vaccines. Given the possible need for vaccine booster shots, however, public health officials should consider proactively acknowledging this possibility to avoid a further loss of trust if or when this happens [15]. Moreover, delivering messages that highlight the benefits of herd immunity could also reduce hesitation about COVID-19 vaccines [33].

With regard to vaccine information sources, a large percentage of participants acquired COVID-19 vaccine information from government agencies and social media. Social media permits the rapid sharing of information and will likely be a promising route for the dissemination of vaccination-related information. However, global trends also indicate that social media has become a platform for anti-vaccine messaging [13]. Adequate immunization program communications via social media are needed to counter vaccine misinformation, especially among minority groups [34].

Because the Chinese government has implemented very timely and effective containment measures since the outbreak of the COVID-19, we found that nearly 90% of participants perceived a low or very low risk of COVID-19 infection, and more than 80% of participants didn’t experience a local rebound of epidemic. Most people had a reduced perception of the destructive impact of the pandemic and therefore didn’t realize the importance of getting vaccinated, which may partly explain the relatively high rate of vaccine hesitancy among the Chinese population. To be specific, people who experience epidemic rebound and quarantine are faced with enormous disruptions to work and daily life [35,36], thus making them have an urge to get back to normal sooner via vaccination, resulting in their lower vaccine resistance and hesitancy. The present findings also showed a decreasing trend of vaccine hesitancy and resistance over time, which could be partially explained by the greater public perceived risk of COVID-19 infection because of the appearance of highly contagious virus strains worldwide. The prevalence of a perceived high risk of infection increased from 8.7% during 29 January–28 February 2021, to 10.3% during 1–26 April 2021 (*p* < 0.001). Although COVID-19 immunization programs have made progress, the public should still be informed that they have a high chance of being infected if they are not vaccinated. Frontline workers had lower rates of vaccine resistance and hesitancy because of their higher occupational risk, which aligns with a web-based survey among healthcare workers [37]. Our findings also indicate that to improve vaccine confidence and uptake, healthcare professionals need to be trained to address hesitancy. Furthermore, people can acquire more comprehensive information and make a scientific decision through proactively accessing the COVID-19 vaccine information, which means that they are more likely to get vaccinated.

Our findings have three potential implications for vaccine rollout policies in China and other countries with similar situations. First, we found that women, young and middle-aged individuals, urban dwellers, and those with a higher income could be target populations for further education. Second, our findings suggest that proactive messaging should highlight the high efficacy rates and low side effects of COVID-19 vaccines that are currently on the market. Third, government agencies and social media should be the main avenues of delivering messages to these specific Chinese target populations, given that the public generally has a high degree of confidence in the Chinese government, and given that social media can more easily reach younger people and those with higher incomes.

Population attitudes about COVID-19 vaccines will fluctuate with waves of the pandemic, thus necessitating the regular tracking of vaccine willingness during global vaccine roll-outs. The present study had extensive geographic coverage across China and a large sample size. It was conducted when the Chinese government had just started to provide free COVID-19 vaccines for the general population. Thus, our findings fill a research gap with regard to public willingness and determinants of COVID-19 vaccination and could serve as a reference for tailoring vaccination campaigns in other countries.

This study also has limitations. First, this was an online survey that used a convenience sampling method. Although this survey covered extensive geographic regions throughout China and had a large sample size, participants were recruited among internet users who were young and highly educated, thus limiting generalization of the findings. Second, the study was not designed to assess the ease of vaccination, which may also influence vaccine hesitancy and resistance. Future research is needed to highlight access to immunization programs, monitor adverse effects after immunization, and assess the efficacy of vaccination in the long term.

## 5. Conclusions

This large national survey was conducted when the Chinese government had just started to provide free COVID-19 vaccines to the general population. We found that more than half of the participants were willing to get vaccinated, but a substantial proportion of participants were hesitant about vaccination. Relatively few were resistant. Women, young and middle-aged individuals, those with a high income, and city dwellers were more likely to be vaccine hesitant and thus should be target populations for further vaccination campaigns. People who are hesitant or resistant about vaccination could be persuaded by leveraging the public’s relatively high confidence in the Chinese government and by taking advantage of the popularity of social media by delivering specific messaging that focuses on vaccine safety and efficacy.

## Figures and Tables

**Figure 1 vaccines-09-01172-f001:**
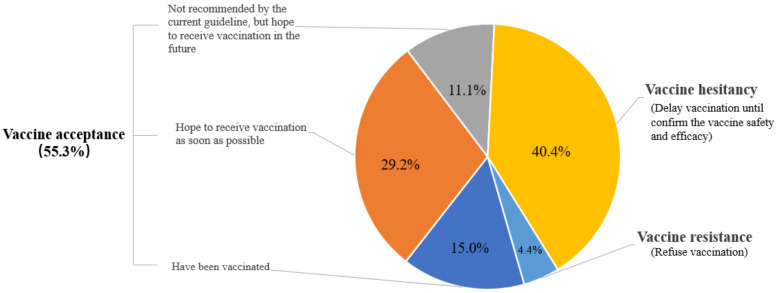
The prevalence of COVID-19 vaccination willingness.

**Figure 2 vaccines-09-01172-f002:**
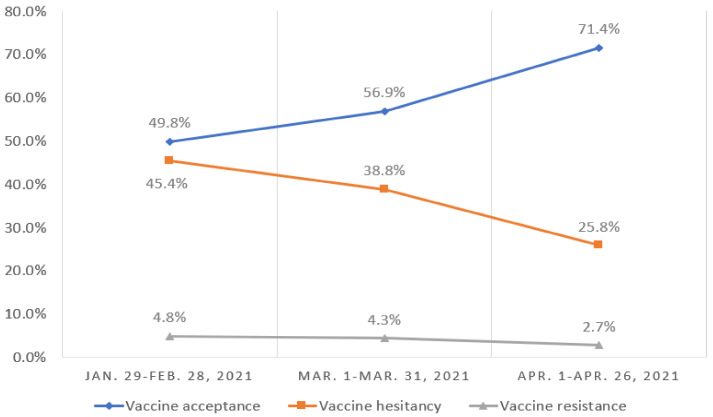
Change of willingness of vaccination against COVID-19 over time.

**Figure 3 vaccines-09-01172-f003:**
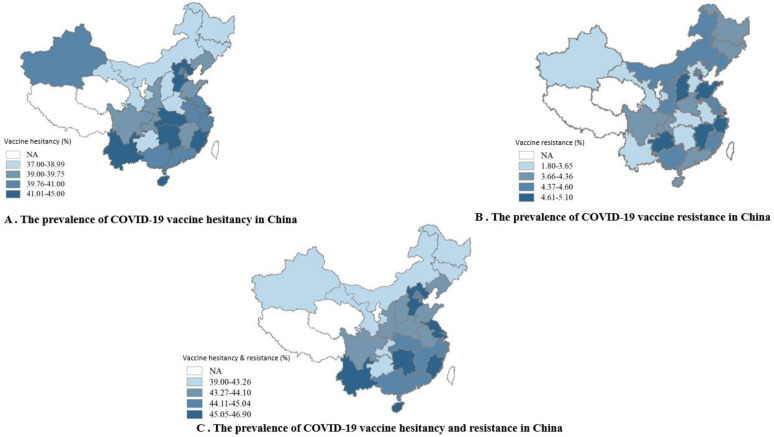
The prevalence of vaccination willingness between provinces in China. NA, not available, the rates of vaccine hesitancy and resistance in Xizang, Qianghai, Ningxia, Hong Kong, Macau, and Taiwan are not presented in the maps because the number of participants from these 6 provinces or regions was less than 46, resulting in the rates not being valid.

**Figure 4 vaccines-09-01172-f004:**
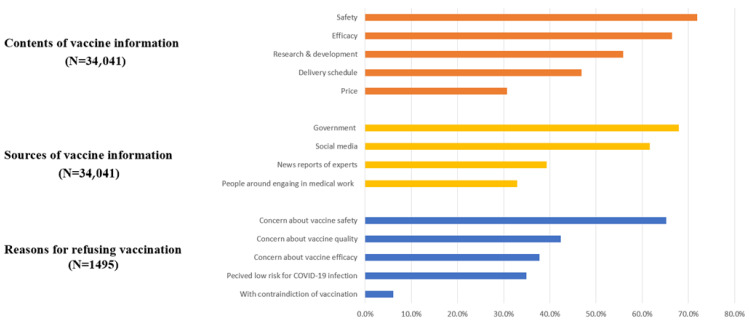
Contents and sources of COVID-19 vaccine information, and reasons for refusing vaccination.

**Table 1 vaccines-09-01172-t001:** Sociodemographic characteristics of the total sample (*n* = 34,041).

Characteristic	Participants, *n* (%)
Gender	
Male	15,732 (46.2)
Female	18,309 (53.8)
Age, years	
18–39	20,727 (60.9)
40–59	12,713 (37.3)
≥60	601 (1.8)
Survey time	
29 January–28 February 2021	13,739 (40.4)
1–31 March 2021	17,396 (51.1)
1–26 April 2021	2906 (8.5)
Geographical region, China ^a^	
Eastern	13,321 (39.2)
Northern	6382 (18.8)
Southern	6064 (17.8)
Central	2965 (8.7)
Northeast	2398 (7.1)
Southwest	1957 (5.8)
Northwest	920 (2.7)
Living area	
Urban	26,942 (79.1)
Rural	7099 (20.9)
Education attainment	
College degree or higher	26,957 (79.2)
Less than college	7084 (20.8)
Marital status	
Married	26,392 (77.5)
Unmarried ^b^	7649 (22.5)
Monthly family income, RMB ^c^	
<5000	8438 (24.8)
5000–11,999	15,961 (46.9)
≥12,000	9642 (28.3)
Frontline worker ^d^	
Yes	1074 (3.2)
No	32,967 (96.8)
COVID-19 infection status	
Confirmed or suspected	104 (0.3)
Not infected	33,937 (99.7)
Experience of quarantine	
Centralized quarantine	493 (1.4)
Home confinement	3388 (10.0)
No	30,160 (88.6)
Local rebound of epidemic	
Yes	6013 (17.7)
No	28,028 (82.3)
History of chronic diseases	
Yes	3103 (9.1)
Unknown	2392 (7.0)
No	28,546 (83.9)
Depression symptoms	
Moderate to severe	3470 (10.2)
Mild	4393 (12.9)
Normal	26,178 (76.9)
Anxiety symptoms	
Moderate to severe	2638 (7.7)
Mild	4555 (13.4)
Normal	26,848 (78.9)

^a^ A total of 34 participants (0.1%) had no data on geographical region. ^b^ The unmarried category included separated, divorced, and widowed. ^c^ As of 24 June 2021, 1 RMB = USD $0.15. ^d^ Frontline worker included healthcare worker, community epidemic prevention personnel, public transport driver, customs officer, and logistics personnel.

**Table 2 vaccines-09-01172-t002:** Sociodemographic characteristics, epidemic-related factors, vaccination-specific factors, and health status of respondents by COVID-19 vaccination willingness.

Characteristic	Intent to Be Vaccinated, *n* (%)/Mean (SD)			*p*
	Vaccine Acceptance (*n* = 18,810)	Vaccine Hesitancy (*n* = 13,736)	Vaccine Resistance (*n* = 1495)	
**Sociodemographic Factor**				
Sex				<0.001
Male	9110 (57.9)	5886 (37.4) ^a^	736 (4.7)	
Female	9700 (53.0)	7850 (42.9)	759 (4.1)	
Age, years				<0.001
18–39	11,264 (54.3)	8591 (41.4) ^a^	872 (4.2)	
40–59	7158 (56.3)	4967 (39.1)	588 (4.6)	
≥60	388 (64.6)	178 (29.6)	35 (5.8)	
Survey time				<0.001
29 January–28 February 2021	6840 (49.8)	6236 (45.4) ^a^	663 (4.8) ^b^	
1–31 March 2021	9894 (56.9)	6749 (38.8)	753 (4.3)	
1– 26 April 2021	2076 (71.4)	751 (25.8)	79 (2.7)	
Geographical region, China				0.421
Eastern	7275 (54.6)	5419 (40.7)	627 (4.7)	
Northwest	518 (56.3)	361 (39.2)	41 (4.5)	
Northern	3527 (55.3)	2575 (40.3)	280 (4.4)	
Northeast	1362 (56.8)	933 (38.9)	103 (4.3)	
Southern	3355 (55.3)	2455 (40.5)	254 (4.2)	
Southwest	1103 (56.4)	775 (39.6)	79 (4.0)	
Central	1649 (55.6)	1206 (40.7)	110 (3.7)	
Living area				<0.001
Urban	14,698 (54.6)	11,025 (40.9) ^a^	1219 (4.5) ^b^	
Rural	4112 (57.9)	2711 (38.2)	276 (3.9)	
Education attainment				<0.001
Less than college	4054 (57.2)	2694 (38)^a^	336 (4.7)	
College degree or higher	14,756 (54.7)	11,042 (41)	1159 (4.3)	
Monthly family income, RMB				<0.001
0–4999	4712 (55.8)	3292 (39)	434 (5.1) ^b^	
5000–19,999	8843 (55.4)	6517 (40.8)	601 (3.8)	
≥12,000	5255 (54.5)	3927 (40.7)	460 (4.8)	
Marital status				<0.001
Married	14,736 (55.8)	10,551 (40)^a^	1105 (4.2) ^b^	
Unmarried	4074 (53.3)	3185 (41.6)	390 (5.1)	
Epidemic-related factors				
COVID-19 infection status				0.575
Confirmed or suspected case	62 (59.6)	39 (37.5)	3 (2.9)	
Not infected	18,748 (55.2)	13,697 (40.4)	1492 (4.4)	
Frontline worker				<0.001
Yes	794 (73.9)	250 (23.3) ^a^	30 (2.8) ^b^	
No	18,016 (54.6)	13,486 (40.9)	1465 (4.4)	
Local rebound of epidemic				0.069
Yes	3312 (55.1)	2468 (41)	233 (3.9)	
No	15,498 (55.3)	11,268 (40.2)	1262 (4.5)	
Quarantine experience				<0.001
Centralized quarantine	347 (70.4)	131 (26.6) ^a^	15 (3.0) ^b^	
Home confinement	2075 (61.2)	1213 (35.8)	100 (3.0)	
None	16,388 (54.3)	12,392 (41.1)	1380 (4.6)	
Perceived risk of infection				<0.001
Low or very low	16,923 (55.3)	12,328 (40.3) ^a^	1351 (4.4)	
Medium	1567 (53.0)	1275 (43.2)	112 (3.8)	
High or very high	320 (66.0)	133 (27.4)	32 (6.6)	
Perceived effectiveness of protective measures				<0.001
Vaccination	7.6 (2.2)	7.1 (2.0) ^a^	6.0 (2.3) ^b^	
Social distancing	7.3 (2.2)	7.1 (2.1) ^a^	6.7 (2.5) ^b^	
Wear a mask	7.6 (2.1)	7.4 (2.0) ^a^	6.9 (2.4) ^b^	
Maintain personal hygiene	7.7 (2.2)	7.5 (2.1) ^a^	7.1 (2.5) ^b^	
Vaccination-specific factors				
Proactively access COVID-19 vaccine information				<0.001
Yes	16,791 (57.1)	11,668 (39.7) ^a^	937 (3.2) ^b^	
No	2019 (43.5)	2068 (44.5)	558 (12.0)	
History of receiving influenza vaccines				<0.001
Yes	7347 (63.8)	3889 (33.8) ^a^	279 (2.4)	
No	11,463 (50.9)	9847 (43.7)	1216 (5.4)	
History of allergic reactions to other vaccines				<0.001
Yes	2174 (53.7)	1630 (40.2)	246 (6.1) ^b^	
No	16,636 (55.5)	12,106 (40.4)	1249 (4.2)	
Health status				
History of chronic diseases				0.965
Yes	1721 (55.5)	1242 (40)	140 (4.5)	
Unknown	1328 (55.5)	955 (39.9)	109 (4.6)	
No	15,761 (55.2)	11,539 (40.4)	1246 (4.4)	
Depression symptoms				<0.001
Moderate to severe	1903 (54.8)	1344 (38.7) ^a^	223 (6.4) ^b^	
Mild	2293 (52.2)	1894 (43.1)	206 (4.7)	
Normal	14,614 (55.8)	10,498 (40.1)	1066 (4.1)	
Anxiety symptoms				<0.001
Moderate to severe	1483 (56.2)	986 (37.4) ^a^	169 (6.4) ^b^	
Mild	2351 (51.6)	1987 (43.6)	217 (4.8)	
Normal	14,976 (55.8)	10,763 (40.1)	1109 (4.1)	

^a^ There were significant differences between vaccine hesitancy and acceptance groups by multiple comparisons. ^b^ There were significant differences between vaccine resistance and acceptance groups by multiple comparisons.

**Table 3 vaccines-09-01172-t003:** Factors that influence vaccine hesitancy and vaccine resistance according to multinomial logistic regression.

Characteristic	Vaccine Hesitancy vs. Acceptance, OR (95% CI)	Vaccine Resistance vs. Acceptance, OR (95% CI)
**Sociodemographic factors**		
Gender		
Male	1 [Reference]	1 [Reference]
Female	1.21 (1.16–1.27)	0.91 (0.82–1.02)
Age, years		
18–39	1.67 (1.38–2.01)	1.00 (0.69–1.45)
40–59	1.53 (1.27–1.84)	1.18 (0.82–1.71)
≥60	1 [Reference]	1 [Reference]
Living area		
Rural	1 [Reference]	1 [Reference]
Urban	1.13 (1.07–1.20)	1.26 (1.09–1.45)
Monthly family income, RMB		
0–4999	1 [Reference]	1 [Reference]
5000–19,999	1.07 (1.01–1.13)	0.81 (0.71–0.92)
≥12,000	1.10 (1.03–1.17)	1.08 (0.93–1.25)
**Epidemic-related factors**		
Survey time		
29 January–28 February 2021	1 [Reference]	1 [Reference]
1–31 March 2021	0.77 (0.73–0.81)	0.84 (0.75–0.94)
1–26 April 2021	0.41 (0.38–0.45)	0.43 (0.34–0.55)
Frontline worker		
Yes vs. No	0.43 (0.37–0.50)	0.50 (0.34–0.73)
Local rebound of epidemic		
Yes vs. No	1.01 (0.95–1.07)	0.84 (0.73–0.98)
Quarantine experience		
Centralized quarantine	0.52 (0.42–0.64)	0.43 (0.25–0.74)
Home confinement	0.80 (0.74–0.86)	0.59 (0.48–0.74)
None	1 [Reference]	1 [Reference]
Vaccine offers less protection than other protective measures		
Yes vs. No	1.52 (1.45–1.60)	2.22 (1.99–2.48)
Perceived low risk of infection		
Low or very low	1.57 (1.27–1.94)	1.00 (0.67–1.47)
Medium	1.77 (1.41–2.20)	0.82 (0.53–1.25)
High or very high	1 [Reference]	1 [Reference]
**Vaccination-specific factors**		
Proactively access COVID-19 vaccine information		
Yes vs. No	0.74 (0.69–0.79)	0.22 (0.20–0.25)
History of receiving influenza vaccines		
Yes vs. No	0.62 (0.59–0.65)	0.40 (0.35–0.46)
History of allergic reactions to other vaccines		
Yes vs. No	1.07 (0.99–1.15)	1.45 (1.25–1.69)
**Health status**		
History of chronic diseases		
Yes vs. No/Unknown	0.88 (0.81–0.96)	1.00 (0.83–1.21)
Depression symptoms ^a^		
Yes vs. No	1.03 (0.98–1.09)	1.20 (1.06–1.36)

Multinomial logistic regression analyses were performed to identify factors that influence vaccine hesitancy and resistance. Vaccine acceptance was set as the reference category. Adjusted for sociodemographic factors (sex, age, marital status, living area, education, and monthly family income), epidemic-related factors (COVID-19 infection status, frontline worker, local rebound of epidemic, quarantine experience, perceived effectiveness of protective measures, and perceived risk of infection), vaccination-specific factors (proactively access information about COVID-19 vaccines, history of receiving influenza vaccines, history of allergic reactions to other vaccines), and health status (history of chronic diseases, depression symptoms, and anxiety symptoms). ^a^ Mild and moderate to severe depression symptoms were classified as depression symptoms.

## Data Availability

The data for this article will be shared on reasonable request to the corresponding author.

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
