# Peer review of "Public Willingness and Determinants of COVID-19 Vaccination at the Initial Stage of Mass Vaccination in China"

_vaccines, 2021, doi:10.3390/vaccines9101172_

Round 1

Reviewer 1 Report

Estimated Authors,

I've read with great interest the present paper on the SARS-CoV-2 vaccine acceptance among residents of Mainland China. The present study identified a relatively low vaccine acceptance, with a significant share of the participants exhibiting either vaccine hesitancy or vaccine resistance. The main drivers of the aforementioned attitutudes are consistent with previous international reports, and collectively stress how difficult will be achieving immunization rates able to truly stop the spreading of this pathogen.

In fact, I've only a couple of minor concerns regarding the present paper, that unfortunately slow down its eventual acceptance. For instance, the English text (particularly in the introduction section) requires some significant intervention, as some typos and awkward sentences penalize the otherwise more than sufficient quality of the text.

Moreover, I would suggest the authors to discuss more extensively about the characteristics of the sample. Large as it was, its representitivity in confront of the general population is very important in order to achieve a fully generalizability of the results. This is even more important as (a common shortcoming of all web-based studies) a self-selection of the paper cannot be ruled out.

Eventually, I would suggest the Authors to discuss about the potential impact of the very effective containment measures that PRC has implemented since the earlier stages of the pandemic. Having reduced the potential impact on the whole of the Chinese population, would it be reasonable that the vaccine hesitancy may have been influenced by a reduced perception of the distructive impact of SARS-CoV-2 on social and economic issues, and not only regarding the health ones?

Author Response

 Response to Reviewer 1 Comments
Point 1: In fact, I've only a couple of minor concerns regarding the present paper, that
unfortunately slow down its eventual acceptance. For instance, the English text
(particularly in the introduction section) requires some significant intervention, as some
typos and awkward sentences penalize the otherwise more than sufficient quality of the
text.
Response 1: We thank the reviewer for pointing out this issue. We have revised and
polished the language of the article, particularly in the introduction section (Line 57-
114).
Point 2: Moreover, I would suggest the authors to discuss more extensively about the
characteristics of the sample. Large as it was, its representitivity in confront of the
general population is very important in order to achieve a fully generalizability of the
results. This is even more important as (a common shortcoming of all web-based
studies) a self-selection of the paper cannot be ruled out.
Response 2: We thank the reviewer for pointing out this issue. Five variables
(experience of quarantine, local rebound of epidemic, history of chronic diseases,
depression symptoms and anxiety symptoms) were added to the previous
sociodemographic characteristics of the total sample, presented by Table 1.
Point 3: Eventually, I would suggest the Authors to discuss about the potential impact
of the very effective containment measures that PRC has implemented since the earlier
stages of the pandemic. Having reduced the potential impact on the whole of the
Chinese population, would it be reasonable that the vaccine hesitancy may have been
influenced by a reduced perception of the distructive impact of SARS-CoV-2 on social
and economic issues, and not only regarding the health ones?
Response 3: We thank the reviewer for pointing out this issue and have revised this
accordingly in the discussion part (Line 360-369).

Reviewer 2 Report

This manuscript is well presented, and clearly structured regarding results, graphs and how their linked to the conclusions;

Limitations of the study, expresed by the authors ending the discussion are reasonable and well defined.

Only to suggest, if possible, to define which could be the "proactive messaging" expressed in the discussion -line 376-. This second finding is discussed in just a sentence and in my opinion could be of interest an extended development.

I recommend its publication, finding this work of interest.

Author Response

 Response to Reviewer 2 Comments
Point 1: Only to suggest, if possible, to define which could be the "proactive
messaging" expressed in the discussion -line 376-. This second finding is discussed in
just a sentence and in my opinion could be of interest an extended development.
Response 1: We thank the reviewer for pointing out this issue and have revised this
accordingly in the discussion part (line 384-387).

Reviewer 3 Report

This article is well written and complete, I only have a few minor comments to make directly in the PDF

Author Response

 Response to Reviewer 3 Comments
Point 1: No need a running title.
Response 1: We thank the reviewer for pointing out this issue and have deleted the
running title.
Point 2: This abstract is too long, it must be made more concise.
Response 2: We thank the reviewer for pointing out this issue and have revised this
accordingly in the abstract part (Line 32-55).
Point 3: You could add the completed STROBE checklist in appendix
Response 3: We thank the reviewer for pointing out this issue and have provided
STROBE checklist in appendix.
Point 4: If we have the 95% CI, we don’t need the p value, which give the same
information.
Response 4: We thank the reviewer for pointing out this issue and have deleted p value
in the Table 3.
Point 5: You could explain in the method section the choice of your reference
categories for the multivariate analysis.
Response 5: We thank the reviewer for pointing out this issue. The reviewer may not
have seen our expression of “with the vaccine acceptance group set as the reference
category” that we stated our choice of the reference categories for the multinomial
logistic regression analyses. We also have revised this accordingly in Table 3.